# AC-ODM: Actor–Critic Online Data Mixing for Sample-Efficient LLM Pretraining

Jing Ma [* 1]   Chenhao Dang [* 2 3 4]   Mingjie Liao [5]

## Abstract

Optimizing pretraining data composition is pivotal for LLM generalization. While dynamic mixing outperforms static strategies by capturing evolving training dynamics, current methods fail to reconcile computational efficiency with sample efficiency and structural flexibility for diverse pipelines. We introduce **Actor–Critic Online Data Mixing (AC-ODM)**, which approaches data mixing from a reinforcement learning perspective with a parameterized policy that we theoretically prove to act as a dynamic linear surrogate maximizing the constructive interference of gradients. To enhance practical flexibility, AC-ODM supports two operational modes: (i) a **proxy mode** for fixed, pre-prepared corpora, where a policy learned on a small model is transferred to a larger target; and (ii) a **non-proxy mode** for direct end-to-end training from scratch without priors. Empirically, AC-ODM significantly outperforms prior methods in convergence speed and downstream accuracy across various architectures. On Pythia-1B, it reaches optimal validation perplexity using up to 66% fewer training steps than competitive baselines, delivering a 27.5% relative improvement in MMLU accuracy and a 2.23× higher pass@1 on HumanEval, all while incurring a virtually negligible ( 0.4%) per-step wall-clock increase and only 2% additional memory overhead. Code is available at https://github.com/DANG-ai/AC-ODM.

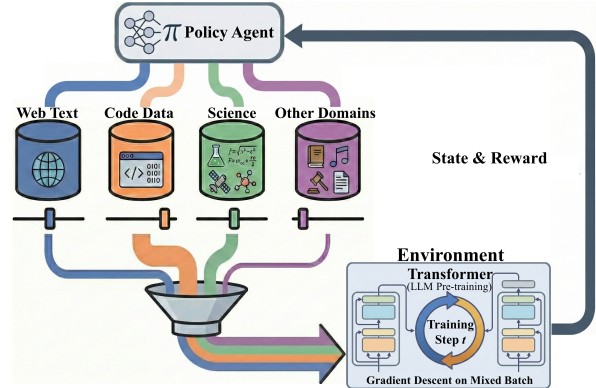

*Figure 1.* **The framework of AC-ODM.** We approach LLM pretraining data mixing from a reinforcement learning perspective. Treating the LLM as the environment, an Actor-Critic agent dynamically senses the training state and adjusts domain sampling weights to explicitly maximize the constructive interference of gradients.

## 1. Introduction

The generalization capability of Large Language Models (LLMs) is intrinsically governed by the quality and distribution of their pretraining corpora. Beyond mere scale, the coverage and mixture of data domains strongly influence sample efficiency, convergence speed, and downstream accuracy (St. John & Draper, 1975; Du et al., 2022; Lee et al., 2022). Consequently, optimizing data mixing has emerged as a critical frontier in efficient LLM pretraining.

Prior research on data mixing has largely focused on *static* strategies, where domain weights are determined offline before training begins. Representative approaches like DoReMi (Xie et al., 2023), DoGE (Fan et al., 2024), RegMix (Liu et al., 2025b), and CHAMELEON (Xie et al., 2025) utilize small proxy models or heuristic leverage scores to estimate global domain importance. While effective, these static weights fail to adapt to the changing learning dynamics of the model during the extensive pretraining process. Recently, research has increasingly shifted toward *dynamic* data mixing, exemplified by methods such as ODM (Albalak et al., 2023) and PiKE (Li et al., 2025). By adjusting data distributions on-the-fly, dynamic strategies generally demonstrate superior effectiveness compared to static baselines, as

---

[*]Equal contribution  [1]BRAIN, Renmin University of China, Beijing, China [2]Shanghai Jiaotong University, Shanghai, China [3]Shanghai Artificial Intelligence Laboratory, Shanghai, China [4]China Electronics Technology Group Corporation 15th Research Institute, Beijing, China [5]LiblibAI, Beijing, China. Correspondence to: Mingjie Liao <liaomingjie@liblib.ai>.

*Proceedings of the $43^{rd}$ International Conference on Machine Learning*, Seoul, South Korea. PMLR 306, 2026. Copyright 2026 by the author(s).

they can respond to the model's evolving capabilities and deficits. However, a critical challenge remains: existing dynamic methods often lack a unified framework that balances computational efficiency with sample efficiency and structural flexibility. For instance, sophisticated selection algorithms may incur prohibitive runtime overhead, while lightweight heuristics may struggle to accommodate diverse pipelines, such as those involving direct pretraining from scratch without priors versus fixed, pre-prepared corpora.

To address these limitations, we propose **Actor–Critic Online Data Mixing (AC-ODM)**, a framework that approaches data mixing from a reinforcement learning (RL) perspective. As illustrated in Figure 1, we treat the LLM pretraining process as an environment where a parameterized policy (the Actor) dynamically optimizes domain weights based on the model's real-time state. Unlike previous heuristics, our approach is grounded in optimization geometry. Theoretically, we prove that the learned policy acts as a dynamic linear surrogate that maximizes the constructive interference of gradients, thereby explicitly optimizing the effective descent magnitude during pretraining. AC-ODM is designed with practical flexibility at its core, supporting two operational modes: a *Proxy Mode* for scenarios with fixed corpora, where a policy is learned on a small model and transferred to guide a larger target; and a *Non-Proxy Mode* for end-to-end training where new domains may emerge dynamically.

---

**Main Contributions of AC-ODM**

- **RL-Based Framework:** We propose AC-ODM, an online data mixing method that uses an Actor-Critic network to capture intra-domain interactions via a novel gradient alignment reward, steering pretraining toward faster convergence.

- **Theoretical Analysis:** We provide a theoretical analysis demonstrating that our reward mechanism acts as a first-order proxy for spectral coherence in the optimization landscape, explicitly maximizing the effective gradient update magnitude.

- **Dual-Mode Flexibility & Superior Performance:** We introduce two complementary modes (Proxy/Non-Proxy) to handle diverse data constraints. Empirically, AC-ODM-410M reaches optimal validation perplexity on Pythia-1B using **66%** fewer steps than baselines and achieves a **27.5%** gain in MMLU accuracy, all while maintaining the fastest convergence on LLaMA3 architectures with virtually negligible computational overhead (<0.5% time and 2% memory).

---

## 2. Related Work

**Data Mixing in LLM Pretraining.** The composition of pretraining data is a primary driver of LLM generalization and sample efficiency, often outweighing pure data volume (St. John & Draper, 1975; Du et al., 2022; Lee et al., 2022; Sorscher et al., 2023; Albalak et al., 2024). Recent technical reports like Qwen3 (Yang et al., 2025) further emphasize that sophisticated mixing strategies are essential prerequisites for training competitive foundation models.

**Static Data Mixing Strategies.** Static approaches determine weights offline. Standard paradigms include training proxy models to minimize loss gaps or align gradients (Xie et al., 2023; Fan et al., 2024). Others leverage scaling laws and regression to predict optimal mixtures (Liu et al., 2025b; Shukor et al., 2025; Yen et al., 2025), or utilize clustering and multi-dimensional quality assessments (Xie et al., 2025; Diao et al., 2025; Zhuang et al., 2025). Despite their diversity, these static strategies inherently fail to adapt to the target model's evolving training dynamics, often resulting in sub-optimal performance compared to dynamic approaches (Li et al., 2025).

**Dynamic Data Mixing Strategies.** To capture feature learning evolution, dynamic methods adjust mixtures on-the-fly. Following the bandit-based ODM (Albalak et al., 2023), recent works incorporate gradient interactions (Li et al., 2025), bi-level optimization (Yu et al., 2025), quality-diversity balance (Liu et al., 2025a), and Bayesian optimization (Ouyang et al., 2025). However, these methods often face a trade-off between computational overhead and structural flexibility, lacking a unified framework to efficiently handle both fixed-corpus and non-prior information scenarios.

## 3. AC-ODM

In this section, we present **Actor-Critic Online Data Mixing (AC-ODM)**, a framework designed for efficient and adaptive pretraining of large language models. We first formulate the problem, then detail the RL-based methodology, followed by a rigorous theoretical analysis of our reward mechanism, and finally describe the operational modes tailored for diverse real-world scenarios.

### 3.1. Problem Formulation

Let $D = \{D_1, \ldots, D_K\}$ be a pretraining corpus composed of $K$ distinct domains. We seek a sequence of domain weights on the probability simplex $\boldsymbol{\alpha} \in \Delta^K \subset \mathbb{R}^K$. Training batches are produced by first sampling a domain $i \sim \boldsymbol{\alpha}$ and then sampling a sequence uniformly within that domain, i.e., $B \sim \text{UNIF}(D_i)$. This induces the instance-wise distribution $P_{\boldsymbol{\alpha}} \triangleq \sum_{i=1}^{K} \alpha_i \cdot \text{UNIF}(D_i)$. While offline data mixing fixes $P_{\boldsymbol{\alpha}}$ before training, AC-ODM updates $P_{\boldsymbol{\alpha}^t}$ at every iteration $t$ to adapt to the model's evolving state, with the objective of maximizing generalization performance while incurring negligible computational overhead.

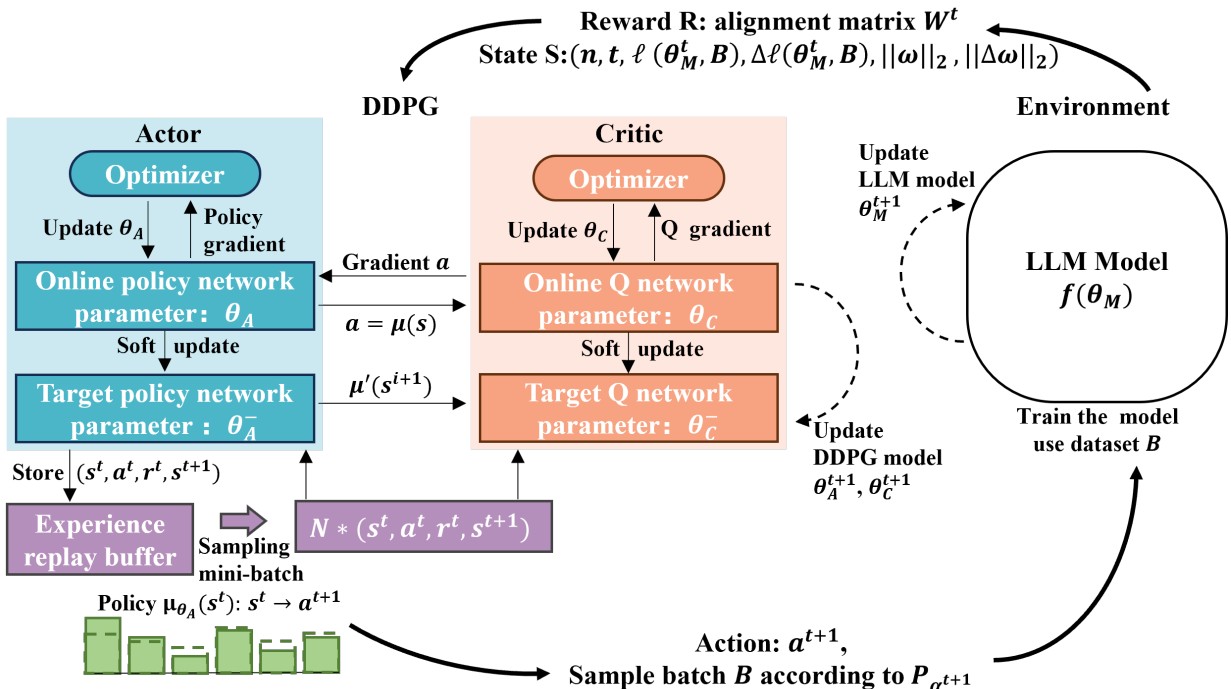

*Figure 2.* **Overview of AC-ODM Framework.** At iteration $t$, the policy $\mu_{\theta_A}$ observes the environment state $s^t$ from the current LLM (e.g., loss dynamics, weight norms) and outputs an action $a^t$ to adjust domain weights $\boldsymbol{\alpha}^t$. A batch $B$ is sampled according to $P_{\boldsymbol{\alpha}^t}$. The loss gradient $\nabla L$ and the gradient alignment matrix $W^t$ are computed to update the LLM parameters $\theta_M$ and generate the reward $r^t$. The transition tuple $(s^t, a^t, r^t, s^{t+1})$ is stored in a replay buffer to update the Actor and Critic networks. This closed-loop feedback explicitly maximizes gradient coherence (see Sec. 3.4).

## 3.2. Adapting Actor-Critic to Online Data Mixing

We cast online data mixing as a continuous control problem within a Markov Decision Process (MDP) and adopt the Deep Deterministic Policy Gradient (DDPG) framework. As illustrated in Figure 2, the LLM itself defines the environment. At each step $t$, the agent observes state $s^t$ and executes an action $a^t = \mu_{\theta_A}(s^t)$ that updates the domain sampling weights $\boldsymbol{\alpha}^t$.

**State Space.** The state $s^t$ must be compact yet informative of the training dynamics. We aggregate observable signals including: the iteration index $t$, the number of samples per domain $n = \{n_i\}_{i=1}^K$, the per-domain loss vector $\ell(\theta_M, B) \in \mathbb{R}^K$ and its step-to-step difference $\Delta\ell$, as well as the $L_2$ norm of selected LLM layer weights $\|\omega\|_2$ and their update magnitude $\|\Delta\omega\|_2$. Formally: $s^t = (n, t, \ell(\theta_M, B), \Delta\ell(\theta_M, B), \|\omega\|_2, \|\Delta\omega\|_2)$.

**Action Space.** The action $a^t \in \mathbb{R}^K$ is mapped to the simplex via a softmax function to produce valid mixing weights $\boldsymbol{\alpha}^{t+1}$. Since both state and action spaces are continuous, DDPG is an ideal optimizer.

## 3.3. Designing the Reward Function

Efficient pretraining requires a reward signal that values data which not only minimizes current loss but also accelerates learning across other domains. We define the reward for domain $i$ based on its **gradient alignment** with the aggregate gradient of the remaining corpus: $W_i \triangleq \langle \nabla\ell_i(\theta_M), \sum_{j\neq i} \nabla\ell_j(\theta_M) \rangle$. This score measures the degree to which an update from domain $i$ is geometrically aligned with the optimization direction of other domains. We denote $W = [W_1, \ldots, W_K]$. To stabilize training, we use an importance-corrected exponential moving average for the final reward: $\hat{r}_i^t = \xi \hat{r}_i^{t-1} + (1 - \xi) \frac{W_i^t}{P_{\alpha_i}^{t-1}}$, where the division by $P_{\alpha_i}^{t-1}$ prevents the policy from collapsing into a trivial solution that only samples already-frequent domains.

## 3.4. Theoretical Analysis: Optimization Geometry and Gradient Coherence

While prior work such as DoGE (Fan et al., 2024) interprets gradient alignment primarily as a statistical predictor for generalization loss, we provide a fundamentally different theoretical grounding based on *optimization geometry*. We demonstrate that maximizing the alignment reward does not merely estimate future performance, but explicitly max-

---

**Algorithm 1** AC-ODM in the non proxy mode

---

**REQUIRE:** $D = \{D_1, \ldots, D_K\}$ grouped data
**REQUIRE:** $\theta_M^0$ target LLM weights, $\theta_A$ actor weights, $\theta_C$ critic weights
**REQUIRE:** $\nabla L_i(\theta_M^t)$ stochastic gradient of $B_i$ at step $t$
**REQUIRE:** Hyperparameters: total steps $T$, step size $\eta^t$, target update coefficient $\tau$, discount factor $\gamma$
 1: Initialize $K = |D|$, set $r_i^0 = 0$ for all $i \in \{1, \ldots, K\}$, initialize critic $Q_{\theta_C}$, actor $\mu_{\theta_A}$, and LLM weights $\theta_M^0$
 2: Copy target networks $\bar{\theta}_C \leftarrow \theta_C, \bar{\theta}_A \leftarrow \theta_A$
 3: Initialize replay buffer $\mathcal{B}$, perform warm up to obtain the initial state $s^0 = (n^0, 0, \ell(\theta_M^0, B), \Delta\ell(\theta_M^0, B), \|\omega^0\|_2, \|\Delta\omega^0\|_2)$
 4: **for** $t = 0$ to $T - 1$ **do**
 5:     Choose action $a^t = \mu_{\theta_A}(s^t)$ and map to domain weights $\alpha^t$
 6:     Sample batch $B^t = \{B_1^t, \ldots, B_K^t\}$ according to $P_\alpha \triangleq \sum_{i=1}^K \alpha_i^t \cdot \mathrm{UNIF}(D_i)$
 7:     Compute $\nabla L_i(\theta_M^t)$ for all $i \in [K]$ and the alignment vector $W^t$
 8:     Update the LLM: $\theta_M^{t+1} \leftarrow \theta_M^t - \eta^t \sum_{i=1}^K \alpha_i^t \nabla L_i(\theta_M^t)$
 9:     Set $r^t \leftarrow W^t$
 10:     Form the next state $s^{t+1} = (n^{t+1}, t+1, \ell(\theta_M^{t+1}, B), \Delta\ell(\theta_M^{t+1}, B), \|\omega^{t+1}\|_2, \|\Delta\omega^{t+1}\|_2)$
 11:     Store $(s^t, a^t, r^t, s^{t+1})$ in $\mathcal{B}$
 12:     Sample $\{(s_k, a_k, r_k, s_k')\}_{k=1}^N$ from $\mathcal{B}$
 13:     Compute $y_k = r_k + \gamma Q_{\bar{\theta}_C}(s_k', \mu_{\bar{\theta}_A}(s_k'))$
 14:     Update critic by minimizing $L = \frac{1}{N}\sum_{k=1}^N (y_k - Q_{\theta_C}(s_k, a_k))^2$
 15:     Update actor via $\nabla_{\theta_A} J \approx \frac{1}{N}\sum_{k=1}^N \nabla_{\theta_A}\mu_{\theta_A}(s_k)\nabla_a Q_{\theta_C}(s_k, a)\big|_{a=\mu_{\theta_A}(s_k)}$
 16:     Soft update targets: $\bar{\theta}_A \leftarrow \tau\theta_A + (1-\tau)\bar{\theta}_A, \bar{\theta}_C \leftarrow \tau\theta_C + (1-\tau)\bar{\theta}_C$
 17: **end for**
 18: **return** actor $\mu_{\bar{\theta}_A}$

---

imizes the **constructive interference** of gradients in the current step, thereby serving as a first-order proxy for the quadratic descent efficiency.

**Setup.** Let $\mathbf{G}^t = [\mathbf{g}_1^t, \ldots, \mathbf{g}_K^t] \in \mathbb{R}^{d \times K}$ denote the gradient matrix at step $t$, where column $\mathbf{g}_i^t$ represents the gradient of domain $i$. The effective update direction of the model is the weighted sum $\mathbf{g}_{total}^t = \mathbf{G}^t \boldsymbol{\alpha}^t$, with $\boldsymbol{\alpha}^t \in \Delta^K$.

**Proposition 1 (Geometric Coherence).** *The AC-ODM reward acts as a linear surrogate for the cross-term energy in the Gram matrix spectrum. Maximizing this reward greedily optimizes the marginal gain in effective gradient magnitude.*

*Proof.* The convergence rate of first-order optimization is dominated by the magnitude of the update vector. We analyze the squared norm of the aggregated gradient using the empirical Gram matrix $\mathbf{H}^t = (\mathbf{G}^t)^\top \mathbf{G}^t \in \mathbb{R}^{K \times K}$:

$$\|\mathbf{g}_{total}^t\|^2 = \|\mathbf{G}^t\boldsymbol{\alpha}^t\|^2 = (\boldsymbol{\alpha}^t)^\top \mathbf{H}^t \boldsymbol{\alpha}^t$$

$$= \underbrace{\sum_{i=1}^K (\alpha_i^t)^2 H_{ii}^t}_{\text{Self-Magnitude}} + \underbrace{\sum_{i \neq j} \alpha_i^t \alpha_j^t H_{ij}^t}_{\text{Interaction Energy}}. \quad (1)$$

Equation (1) reveals that the effective descent magnitude consists of a self-magnitude term and an *interaction energy* term. The interaction energy is determined by the

off-diagonal entries $H_{ij}^t = \langle \mathbf{g}_i^t, \mathbf{g}_j^t \rangle$. Positive entries indicate geometric alignment (constructive interference), while negative entries indicate conflict.

Directly maximizing this quadratic form $(\boldsymbol{\alpha}^t)^\top \mathbf{H}^t \boldsymbol{\alpha}^t$ is computationally expensive. However, the AC-ODM reward for domain $i$, defined as $r_i^t = \langle \mathbf{g}_i^t, \sum_{j \neq i} \mathbf{g}_j^t \rangle$, corresponds to the row-sum of the off-diagonal elements of $\mathbf{H}^t$ (effectively assuming a uniform prior for $\alpha_{j \neq i}$). Consequently, the objective function optimized by the policy, $J(\boldsymbol{\alpha}) = \mathbb{E}_{\boldsymbol{\alpha}}[\mathbf{r}^t] = \sum \alpha_i r_i$, acts as a **linearized surrogate** for the interaction energy. By assigning higher probability mass to domains with high $r_i^t$, the policy steers the optimization trajectory towards regions of maximal spectral coherence, ensuring that the sampled gradients constructively reinforce each other to maximize the effective step size $\|\mathbf{g}_{total}^t\|$. $\square$

### 3.5. Model update

Each iteration updates three parameter sets: the LLM $\theta_M$, the critic $\theta_C$, and the actor $\theta_A$.

**Updating $\theta_M$.** Given $a^t$ and the induced weights $\alpha^t$, we sample $B$ according to $P_\alpha$ and compute the per domain losses and gradients. The proxy model is then updated with

---

**Algorithm 2** AC-ODM in the proxy mode

---

**REQUIRE:** Proxy LLM initialization $\theta_{M,\text{proxy}}^0$, target LLM initialization $\theta_{M,\text{tgt}}^0$, actor $\theta_A$, critic $\theta_C$, domains $D$

1: **Proxy stage:** Train the actor and critic with the proxy LLM using Algorithm 1 on $D$, obtain the trained actor $\mu_{\bar{\theta}_A}$
2: **Transfer:** Freeze the actor and remove reward computation
3: **Target stage:** For steps $t = 0$ to $T_{\text{tgt}} - 1$, sample batches for the target LLM with $\alpha^t = \mu_{\bar{\theta}_A}(s^t)$, update $\theta_{M,\text{tgt}}$ with the reweighted loss, and refresh the state $s^{t+1}$ as in Algorithm 1 without updating the actor or critic
4: **return** target LLM trained under the transferred actor policy

---

a loss reweighting factor $\alpha$:

$$\theta_M^{t+1} \triangleq \theta_M^t - \eta^t \sum_{i \in [k]} \alpha_i^t \nabla \ell_i(\theta_M^t). \qquad (2)$$

**Updating $\theta_C$ and $\theta_A$.** Let the critic be $Q_{\theta_C}(s, a)$ and the actor be $\mu_{\theta_A}(s)$. We compute $r^t = W^t$ and the next state $s^{t+1} = (n^{t+1}, t + 1, \ell(\theta_M^{t+1}, B), \Delta\ell(\theta_M^{t+1}, B), \|\omega^{t+1}\|_2, \|\Delta\omega^{t+1}\|_2)$, then store $(s^t, a^t, r^t, s^{t+1})$ in the replay buffer. For mini batch samples $\{(s_k, a_k, r_k, s_k')\}_{k=1}^N$, the temporal difference target is

$$y_k = r_k + \gamma Q_{\bar{\theta}_C}(s_k', \mu_{\bar{\theta}_A}(s_k')). \qquad (3)$$

The critic minimizes

$$L = \frac{1}{N} \sum_{k=1}^N \big(y_k - Q_{\theta_C}(s_k, a_k)\big)^2. \qquad (4)$$

The actor ascends the policy gradient

$$\nabla_{\theta_A} J \approx \frac{1}{N} \sum_{k=1}^N \nabla_{\theta_A} \mu_{\theta_A}(s_k) \, \nabla_a Q_{\theta_C}(s_k, a)\big|_{a = \mu_{\theta_A}(s_k)}. \qquad (5)$$

We follow DDPG and maintain target networks for stability.

### 3.6. Modes of AC-ODM and Applications

AC-ODM supports two operational modes designed to address different constraints in real-world large-scale training.

**Non-Proxy Mode (End-to-End).** In this mode (Algorithm 1), the actor and critic are co-trained with the target LLM from scratch. *Application Scenario:* This mode is tailored for direct pretraining scenarios where no prior knowledge of domain characteristics or auxiliary proxy models is available. It enables the target LLM to dynamically learn and adjust data mixtures on-the-fly, achieving competitive performance with virtually negligible computational overhead ($< 0.5\%$ wall-clock time).

**Proxy Mode (Policy Transfer).** In this mode (Algorithm 2), the policy is first learned on a small proxy model and then transferred to guide the target LLM. *Application Scenario:* This mode is best suited for standard pretraining pipelines with fixed corpora, where maximizing final downstream

performance is paramount. By decoupling policy learning from target training, it mitigates early-stage exploration noise on the large model. Empirically, this mode yields the strongest generalization, justifying the one-time proxy training cost.

## 4. Experiment

### 4.1. Experimental Setup

We describe datasets, model training protocols, baseline configurations, and evaluation criteria. For the actor and critic networks, we also detail the state design and reward design. All experiments are run on a single machine with an Intel(R) Xeon(R) Platinum 8468 CPU and 8 NVIDIA H800 GPUs with 80 GB memory each.

**LLM training.** We use The Pile Gao et al. (2020), an open source corpus of 825 GB from 22 diverse sources such as YouTube Subtitles, GitHub, and Wikipedia. In addition, we pretrain on SlimPajama (Soboleva et al., 2023), a seven domain corpus containing 672B tokens at a smaller scale. Models are decoder only Transformers implemented with a modified GPT NeoX library Black et al. (2022). Unless noted otherwise, configurations follow Pythia Biderman et al. (2023) and we train a 1 billion parameter model. Each GPU processes a micro batch of 8 sequences. We use gradient accumulation across 8 GPUs with accumulation step 18, which yields an effective batch size of 1152 samples. For each batch, we first draw 10 percent per domain to expose the policy to intra domain relationships while preserving exploration and exploitation. The sequence length is 1024 with sequence packing Roberts et al. (2022). Training runs for 41,667 steps, corresponding to 50 billion tokens. During the first 833 warmup steps we replace AC driven weights with The Pile domain weights perturbed by Gaussian noise sampled from $N(0, 0.02)$.

**AC training.** The actor and critic share the same warmup and main training schedules as the LLM, with cosine decay learning rate starting at 0.01 and decaying to 0.001. During warmup, we train the actor and critic from the replay buffer $\mathcal{B}$. To initialize the actor, we use the LLM warmup domain weights as soft labels, namely the noisy The Pile weights, and optimize with mean squared error. For the critic, we initialize labels as $(1 + \gamma)r^t$ and optimize with mean squared

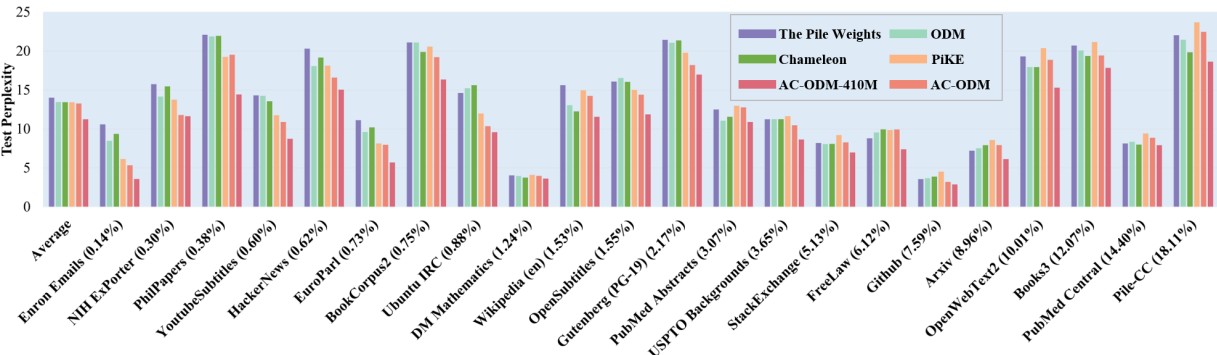

*Figure 3.* **Test Perplexity Breakdown on The Pile.** We report the test perplexity across 22 individual domains. AC-ODM-410M achieves the lowest perplexity in 17 out of 22 domains, showing robust generalization. It effectively balances performance on dominant domains (e.g., Pile-CC) while significantly improving on specialized ones (e.g., DM Mathematics), outperforming both leverage-score based static mixing (CHAMELEON) and gradient-conflict reduction strategies (PiKE).

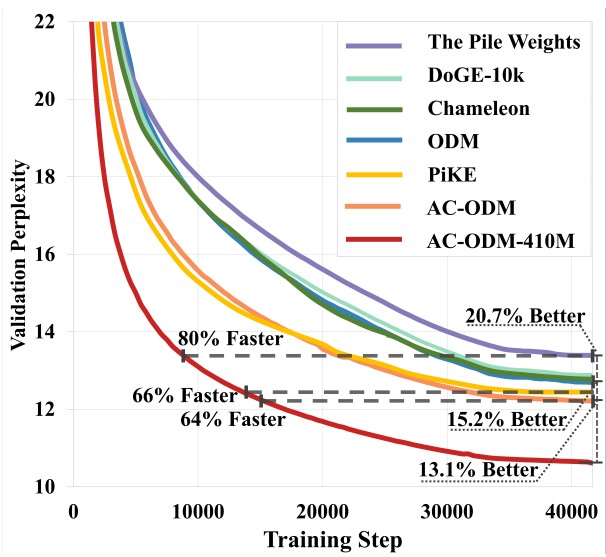

*Figure 4.* **Validation Perplexity on The Pile.** AC-ODM-410M (Proxy) demonstrates the fastest convergence, significantly outperforming static baselines (CHAMELEON, DoReMi) and dynamic methods (PiKE, ODM). It reaches the optimal perplexity of the strongest baseline with 66% fewer steps.

error. During main training, each iteration samples 256 tuples from $\mathcal{B}$, dispatches 32 tuples per GPU, and uses gradient accumulation of 1, which gives an effective batch size of 256. Architectural details for Pythia 1B and the AC networks are in Appendix A.

**Reward and State setting.** For the state features in Pythia 1B, the term $\|\omega\|_2$ is computed on a subset of layers. We use the first Transformer layer together with all layers whose indices are even. This selection reduces computation time with negligible loss of fidelity. To balance efficiency and efficacy in reward computation, we restrict the calculation to a subset of parameters. Specifically, for Pythia 1B we use the final feedforward blocks of Transformer layers 12, 14,

and 16, which together contain 50,331,648 parameters. This choice reduces memory traffic while preserving a faithful proxy for reward estimation. Ablations in Appendix B show that when only three layers are used this selection is optimal.

**Baselines.** We compare AC-ODM against a comprehensive set of baselines representing both static and dynamic paradigms. For static strategies, we evaluate: (1) The Pile Weights (TPW) Gao et al. (2020), the original heuristic mixture; (2) DoReMi Xie et al. (2023), which derives weights via a proxy model; (3) DoGE Fan et al. (2024), which optimizes weights through gradient alignment; and (4) CHAMELEON Xie et al. (2025), which utilizes leverage scores in an embedding space. For dynamic strategies, we compare against: (5) ODM Albalak et al. (2023), which employs a multi-armed bandit algorithm; and (6) PiKE Li et al. (2025), which adapts weights based on gradient conflicts. All baselines are implemented and trained under identical hardware configurations and computational budgets to ensure strict fairness.

**Evaluation.** We report validation and test perplexity averaged over all domains. For downstream generalization, we evaluate on MMLU Hendrycks et al. (2021) with zero shot and five shot settings and on HumanEval Chen et al. (2021) with pass@1. These protocols are applied to models pretrained on The Pile and to models pretrained on SlimPajama under the same training recipe unless noted otherwise. In addition, for models pretrained on The Pile, we evaluate zero shot accuracy on five representative tasks that probe commonsense and scientific reasoning, namely COPA (Roemmele et al., 2011), SciQ (Welbl et al., 2017), LogiQA (Liu et al., 2020), PIQA (Bisk et al., 2020), and WinoGrande (Sakaguchi et al., 2021). Together, these evaluations measure both language modeling quality and transfer to diverse downstream tasks.

**Supplementary Analyses.** Ablation studies concerning

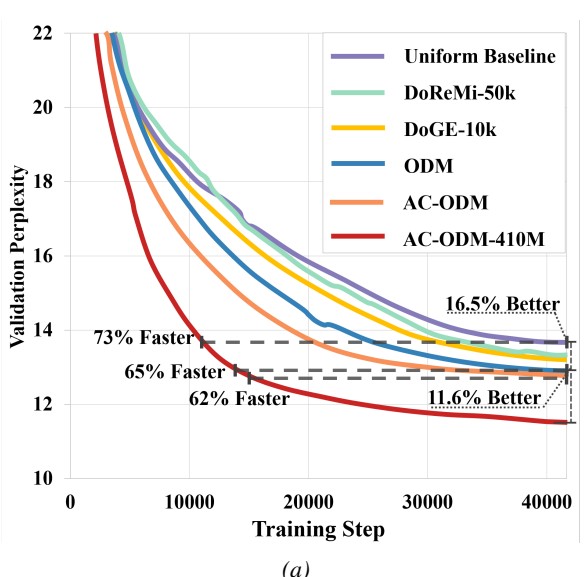
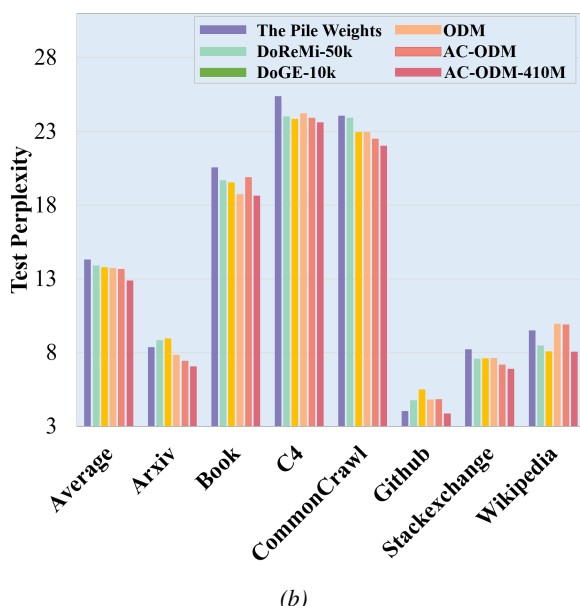

(a)        (b)

*Figure 5.* **Results on SlimPajama with Pythia 1B.** **(a)** Validation perplexity during pretraining. AC-ODM and AC-ODM-410M converge faster than static and online baselines. AC-ODM-410M reaches the best perplexity of ODM in substantially fewer steps and yields lower perplexity at a fixed budget, consistent with the annotations. **(b)** Test perplexity averaged over domains and reported per domain. AC-ODM-410M attains the best average perplexity and is competitive or best across individual domains.

proxy model scaling are presented in Appendix D, followed by an investigation into state components in Appendix E. We visualize the evolution of domain weights during training in Appendix F, and provide a detailed analysis of MMLU task results in Appendix G. Additional camera-ready experiments on policy size, reward stabilization, proxy-target scaling, larger LLaMA-style models, and RegMix are reported in Appendix H.

### 4.2. Main Results

**Convergence Analysis.** Figure 4 presents the validation perplexity on The Pile. The proxy-based AC-ODM-410M exhibits the fastest convergence trajectory, reaching the optimal perplexity of the strongest prior baseline (ODM) with approximately 66% fewer steps. Notably, it surpasses newly introduced strong baselines: it outperforms the static embedding-based method CHAMELEON, confirming that adaptive weights are crucial for capturing evolving training dynamics; it also exceeds the dynamic method PiKE. While PiKE effectively mitigates gradient conflicts, AC-ODM explicitly maximizes constructive interference, leading to more efficient descent directions. At 41,667 steps, AC-ODM-410M achieves lower perplexity than TPW, ODM, and AC-ODM by 20.7%, 16.4%, and 13.1%, respectively. SlimPajama in Figure 5a exhibits the same pattern, where AC-ODM-410M requires 65% fewer steps than ODM and 73% fewer than the uniform baseline to reach its best perplexity and at 41,667 steps improves perplexity by 16.5%

over uniform and 11.6% over the best online baseline. Overall, the proxy mode yields the strongest performance on both corpora, while the non-proxy mode consistently improves over static and online baselines with negligible per-step overhead.

**Domain-Level Generalization.** AC-ODM's reward mechanism explicitly favors domains that generalize well, enabling the policy to exploit shared structures. On The Pile (Figure 3), AC-ODM-410M attains the best average test perplexity and outperforms PiKE in 17 of 22 domains, demonstrating that maximizing gradient alignment is more effective than the conflict reduction of PiKE or the static leverage scores of CHAMELEON. Notably, gains are most pronounced in small and medium domains while remaining significant in dominant ones, indicating that the policy effectively balances within-domain learning with cross-domain transfer. On SlimPajama (Figure 5b), AC-ODM-410M also yields the lowest average perplexity, though with smaller margins than on The Pile due to the coarser granularity of the seven domains limiting complex cross-domain interactions. Together, these results confirm that AC-ODM is particularly advantageous for large, finely partitioned corpora, while consistently improving convergence on coarser collections.

Table 1 summarizes results on MMLU and HumanEval. Consistent with training dynamics, dynamic strategies outperform static ones like Chameleon. Notably, AC-ODM-410M surpasses the strongest baseline PiKE by substan-

*Table 1.* Evaluation of downstream tasks on MMLU and HumanEval. Acc denotes accuracy.

| Algorithm | MMLU 0-s (Acc) | MMLU 5-s (Acc) | HumanEval (p@1) |
|---|---|---|---|
| TPW | 0.20664 | 0.27469 | 0.14119 |
| DoReMi-50k | 0.21862 | 0.27887 | 0.14215 |
| DoGE-10k | 0.22301 | 0.28117 | 0.15651 |
| Chameleon | 0.22065 | 0.28257 | 0.14824 |
| ODM | 0.23514 | 0.28416 | 0.32510 |
| PiKE | 0.24839 | **0.30439** | 0.52188 |
| AC-ODM | 0.25146 | 0.29868 | 0.60256 |
| AC-ODM-410M | **0.29980** | **0.35215** | **0.72644** |

*Note:* **0-s / 5-s**: 0-shot / 5-shot accuracy; **p@1**: pass@1 rate.

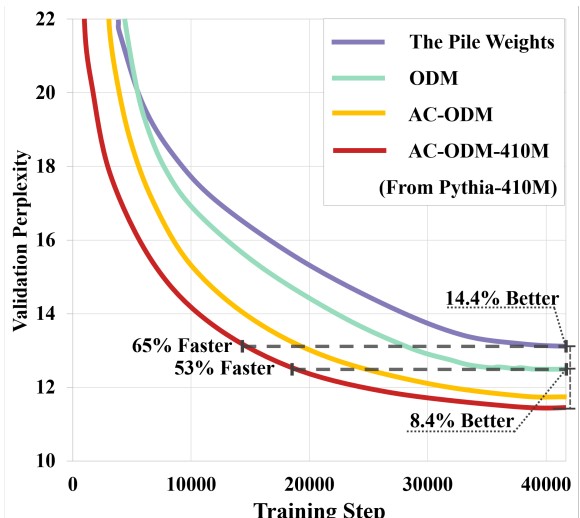

*Figure 6.* Validation perplexity during LLaMA 0.9B pretraining on The Pile. We compare The Pile Weights, ODM, AC-ODM, and AC-ODM-410M, where the latter transfers an actor learned with a 410M Pythia proxy.

tial margins, achieving a $+5.1\%$ gain in zero-shot MMLU and a $+39\%$ relative improvement in HumanEval pass@1, from 0.726 to 0.521. Compared to ODM, AC-ODM-410M improves by $27.5\%$ and $23.9\%$ on zero-shot and five-shot MMLU, and achieves a $2.23\times$ higher pass@1 on HumanEval. Appendix C provides additional zero-shot evaluations on COPA, SciQ, LogiQA, PIQA, and WinoGrande using the same The Pile pretrained checkpoints. These downstream gains are informative because they are not confined to language-modeling perplexity. The improvement on HumanEval is especially large, yet the policy does not simply upweight all code-heavy domains; the domain-weight traces in Appendix F show increases for StackExchange and several high-quality general-purpose domains, while GitHub is reduced. This suggests that AC-ODM improves code generation mainly through better global optimization and transferable reasoning signals, with domain reweighting acting as a mechanism rather than a narrow task-specific shortcut.

*Table 2.* Model size and computational cost during pretraining. Columns AC, LLM, and Total denote parameter counts. AC-ODM(160M/410M) rows report steps to converge the proxy policy, while AC-ODM(1B) rows report steps for the target to match ODM perplexity.

| Method | AC | LLM | Total | Time (s) | Steps | Speedup |
|---|---|---|---|---|---|---|
| ODM | 0 | 1B | 1B | **2.47** | 41667 | 1.00× |
| PiKE | 0 | 1B | 1B | 2.53 | 31250 | 1.30× |
| AC-ODM | 17M | 1B | 1.02B | **2.48** | 28356 | 1.46× |
| AC-ODM(160M) | 17M | 160M | 177M | 0.65 | 28690 | 2.08× |
| Target (1B) | 17M | 1B | 1.02B | 2.48 | 12500 | |
| AC-ODM(410M) | 17M | 410M | 427M | 1.41 | 28690 | 1.47× |
| Target (1B) | 17M | 1B | 1.02B | 2.48 | 12010 | |

*Note:* **Time (s)** for PiKE represents the global average per-step time due to its variable overhead. All rows use Pythia-family target/proxy models; "Target (1B)" denotes the transfer phase. On the LLaMA-style 0.9B target, non-proxy AC-ODM gives about $1.44\times$ speedup.

### 4.3. Generalization to LLaMA-style Architectures

To assess whether AC-ODM extends beyond Pythia, we repeat the pretraining study on a LLaMA-style decoder-only Transformer (Dubey et al., 2024) with 0.9B parameters. As shown in Figure 6, AC-ODM improves the training dynamics of this modern architecture in the same way as for Pythia. The proxy mode remains the strongest: it reaches a target validation perplexity with substantially fewer steps and achieves lower perplexity at a fixed budget. In particular, the annotations in Figure 6 indicate that AC-ODM-410M reduces the steps required to match a common perplexity level by about $65\%$ relative to The Pile Weights and by about $53\%$ relative to AC-ODM, and at the 41,667 step budget it improves perplexity by $14.4\%$ over The Pile Weights and by $8.4\%$ over AC-ODM. The relative margin on the LLaMA-style model is smaller than on Pythia, which is expected because stronger dense decoder designs leave less optimization slack for data mixing alone to exploit. Importantly, the direction of the gain is unchanged: the transferred policy still improves both sample efficiency and final perplexity. This indicates that AC-ODM is not tied to a particular GPT-NeoX-style architecture, but instead leverages gradient relations that persist across decoder families.

### 4.4. Computational cost

We compare the computational resources required by AC-ODM, ODM, and PiKE to train a 1B LLM to the validation perplexity achieved by ODM under identical hardware.

**Per-step Efficiency.** Direct AC-ODM demonstrates high efficiency, incurring a minimal $0.4\%$ overhead per step (2.48 s compared to 2.47 s for ODM). In contrast, **PiKE** exhibits a notably higher latency of 2.53 s per step, attributed to the computational cost of estimating gradient conflicts.

**End-to-End Speedup.** AC-ODM reduces the total training steps by 31.95% (from 41,667 to 28,356), resulting in a $1.46\times$ end-to-end speedup, which effectively surpasses the $1.30\times$ speedup achieved by PiKE.

In the proxy mode, the actor–critic is learned on a smaller proxy and then transferred to the 1B target, which subsequently requires only 28.82% of the ODM steps (12,500 or 12,010 steps). Even when accounting for the proxy stage, the overall speedup reaches $2.08\times$ with a 160M proxy and $1.47\times$ with a 410M proxy. Although using a larger proxy increases the pretraining cost, the stronger policy it acquires amortizes effectively over larger targets, rendering the proxy mode increasingly attractive at scale. These results reveal a practical trade-off between exploration cost and target-stage savings. Non-proxy AC-ODM is preferable when no proxy run is available or when corpora change online, because its per-step overhead is essentially indistinguishable from ODM. Proxy AC-ODM is preferable when the data mixture is fixed and the target model is expensive: the one-time policy-learning cost is paid on a much smaller model, while the target benefits from a mature policy from the first update. Thus the two modes are complementary rather than competing operating points.

### 4.5. Effect of Domain Granularity

AC-ODM benefits from domain partitions that expose meaningful cross-domain gradient structure. To assess this sensitivity, we merge the 22 Pile domains into 11 and 5 semantically related groups and rerun non-proxy AC-ODM on Pythia-1B for 25B tokens. Table 3 shows that coarser partitions consistently weaken validation perplexity, especially in early and middle training. This trend explains why gains are larger on The Pile than on the seven-domain SlimPajama corpus, and suggests that AC-ODM is most informative when domains are sufficiently distinct rather than heavily redundant. The degradation is monotonic across the tested granularity levels, indicating that the actor benefits from observing separable sources of transfer rather than aggregated buckets. When related domains are merged, positive and negative interactions can cancel inside the same group, making the reward less discriminative. In practice, this means AC-ODM should be paired with domain taxonomies that preserve meaningful differences in style, knowledge, and supervision signal; overly coarse taxonomies remain usable, but they reduce the policy's ability to discover fine-grained curricula.

## 5. Limitations

AC-ODM is designed for corpora that can be organized into meaningful domains, and its gains are naturally strongest when this partition exposes useful cross-domain gradient structure. As suggested by the granularity study, coarser

*Table 3.* Effect of domain granularity on AC-ODM. We report validation perplexity for non-proxy Pythia-1B training on merged Pile partitions.

| # Domains | 5,208 | 10,416 | 15,624 | 20,832 |
|---|---|---|---|---|
| 22 | 18.05 | 15.44 | 14.25 | **13.43** |
| 11 | 18.96 | 15.82 | 14.41 | 13.85 |
| 5 | 19.13 | 16.19 | 14.83 | 14.09 |

or highly overlapping groupings remain usable but may provide a less discriminative reward signal. The proxy mode also relies on the learned policy transferring from a smaller model to the target model; our experiments support this behavior across the studied settings, while future work could further examine broader architecture and corpus shifts. Finally, to keep the method lightweight, the reward is estimated from selected parameters and optimizes the mixture of available data rather than the intrinsic quality or governance properties of the data itself. AC-ODM is therefore best viewed as a complementary component to careful data curation, filtering, and auditing in practical pretraining pipelines.

## 6. Conclusion

In this work, we established **Actor-Critic Online Data Mixing (AC-ODM)** as a rigorous framework that reformulates pretraining data selection from a heuristic process into a principled reinforcement learning problem. By theoretically grounding the reward signal in optimization geometry, AC-ODM explicitly maximizes the spectral coherence of updates, ensuring that data mixtures constructively interfere to accelerate convergence rather than merely reducing conflicts. Crucially, our framework reconciles the tension between computational efficiency and structural flexibility through its dual operational modes, seamlessly accommodating both proxy-based transfer and direct pretraining from scratch. Empirical evaluations confirm that AC-ODM sets a new state-of-the-art, outperforming strong dynamic baselines like PiKE with negligible wall-clock overhead ($< 0.5\%$) while reducing the training steps required for optimal perplexity by **66%**. With substantial gains in downstream reasoning (27.5% on MMLU) and code generation ($2.23\times$ on HumanEval), AC-ODM demonstrates that highly efficient data mixing, delivering both rapid convergence and superior sample utilization, is decisive for cultivating superior foundation models.

## Acknowledgements

This project was supported by the National Natural Science Foundation of China (Grant Nos. 62272466, U24A20233, and 12571301).

## Impact Statement

This paper presents a reinforcement learning-based method for optimizing pretraining data mixtures for large language models. Its primary anticipated benefit is improved sample efficiency: by reaching comparable or better model quality with fewer training steps, AC-ODM can reduce the compute, energy use, and carbon emissions associated with large-scale pretraining. At the same time, more efficient pretraining may lower the barrier to developing capable language models, which can amplify both beneficial applications and familiar risks of LLM deployment, including misuse, biased outputs, and uneven access to model-building infrastructure. These risks are not unique to our method, but they make transparent reporting of data composition, evaluation protocols, and deployment safeguards important when applying AC-ODM in practice.

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

*Table 4.* Ablation study of selected layers used for reward computation.

| Block indexes | Perplexity |
|---|---|
| 12, 14, 16 | 13.0655 |
| 14, 15, 16 | 13.0682 |
| 6, 8, 10 | 13.0709 |
| 1, 2, 3 | 13.0701 |

*Table 5.* Zero shot accuracy on downstream tasks using The Pile pretrained 1B models. AVG is the macro average across tasks.

| Task | TPW | DoReMi-50k | DoGE-10k | ODM | AC-ODM | AC-ODM-410M |
|---|---|---|---|---|---|---|
| MMLU | 0.27469 | 0.27887 | 0.27955 | 0.28416 | 0.29868 | **0.35215** |
| COPA | 0.54800 | 0.62000 | 0.64800 | 0.68000 | 0.69800 | **0.72000** |
| SciQ | 0.62000 | 0.63800 | 0.66400 | 0.68900 | 0.70200 | **0.73000** |
| LogiQA | 0.23810 | 0.24580 | 0.27650 | 0.30720 | 0.30110 | **0.32260** |
| PIQA | 0.60330 | 0.61670 | 0.62000 | 0.68330 | 0.69670 | **0.72000** |
| WinoGrande | 0.50930 | 0.52630 | 0.53200 | 0.59650 | 0.58300 | **0.63380** |
| AVG | 0.50374 | 0.52936 | 0.54810 | 0.59120 | 0.59616 | **0.62528** |

## A. Model Configuration

### A.1. LLM Model Configuration

We adopt the sequence length of 1024 and employ a 16-layer Transformers architecture with a hidden size of 2048 and 16 attention heads. Rotary positional embedding Su et al. (2023) is incorporated. We leverage FlashAttention Dao et al. (2022), which optimizes memory access and reduces computation overhead, to improve training efficiency. The model is trained using Adam optimizer Kingma & Ba (2017). The learning rate undergoes a linear warm-up for 833 iterations, gradually increasing from a minimum of 2.5e-5 to a peak of 2.5e-4, followed by a cosine decay back to 2.5e-5. We utilize the GPT-NeoX-20B tokenizer Black et al. (2022) for text processing.

### A.2. AC Networks Configuration

For both the actor and the critic, we employ a fully connected 6-layer neural network with 1024 neurons per hidden layer. Except for the output layer, each layer is followed by layer normalization and a ReLU activation. In the actor, the output layer is further processed by the softmax activation function, while in the critic, the output layer is post-processed by the identity activation function.

## B. Ablation study of selected layers

The results show that using later Transformer blocks yields the best proxy for reward computation: selecting layers 12,14,16 attains the lowest perplexity, slightly outperforming contiguous later layers 14,15,16 and clearly matching or exceeding mid and early layer choices. Although the absolute differences are small, they are consistent, suggesting that mid-to-late representations provide a more informative signal while confirming that AC-ODM is robust to the exact layer subset. These findings support our default choice of 12,14,16.

## C. Zero shot accuracy on downstream tasks

**Analysis.** AC-ODM-410M achieves the highest accuracy on every task and the best average (0.62528), improving over ODM by an absolute +0.0341 and a relative +5.8%. Gains are consistent across commonsense and reasoning benchmarks, with the largest jump on MMLU. The non proxy AC-ODM also improves over ODM on average but trails the proxy mode, underscoring the benefit of learning the policy with a proxy model before guiding the target LLM.

*Table 6.* Ablation study of state components used by AC-ODM. Removing any component degrades performance; "Impr." is the relative change in perplexity compared with using all components.

| Status of state | Perplexity | Impr. |
|---|---|---|
| All components | 13.0655 | – |
| w/o $n$: number of samples per domain | 13.1203 | $-0.419\%$ |
| w/o $t$: iteration step | 13.0958 | $-0.232\%$ |
| w/o $\ell(\theta_M, B)$: per-domain losses | 13.8992 | $-6.38\%$ |
| w/o $\Delta\ell(\theta_M, B)$: change of per-domain losses | 13.5470 | $-3.69\%$ |
| w/o $\|\omega\|_2$: $L^2$ norm of selected layer weights | 13.9115 | $-6.48\%$ |
| w/o $\|\Delta\omega\|_2$: $L^2$ norm change of selected layers | 13.4254 | $-2.75\%$ |

## D. subEffect of proxy model size

Figure 7 compares a 1B target trained with sampling policies learned from 70M, 160M, and 410M proxy LLMs against joint training with AC-ODM. The proxies attain training losses of 2.8, 2.65, and 2.48, with validation perplexities 20.3, 15.5, and 12.1, respectively. Policies from 160M and 410M consistently outperform joint AC-ODM, indicating that a prelearned actor adapts from the first step, whereas an online actor is still converging. The 70M proxy performs worst, suggesting insufficient capacity to learn a transferable policy. The 160M proxy nearly matches the 410M proxy, especially early in training, likely because the 1B target limits headroom. We expect the gap to widen for larger targets and leave a systematic study of proxy–target scaling for future work.

## E. Ablation study of state components

**Analysis.** All six features contribute to policy quality. Removing per-domain losses $\ell(\theta_M, B)$ or the weight norm $\|\omega\|_2$ causes the largest degradations ($\approx 6.4\%$), indicating that absolute training signal and model-scale dynamics are critical for the actor. The change-of-loss term $\Delta\ell(\theta_M, B)$ is also important ($-3.69\%$), while the count of seen samples $n$ and the step index $t$ provide smaller but nontrivial gains. Overall, the full state offers the best perplexity and each component carries complementary information.

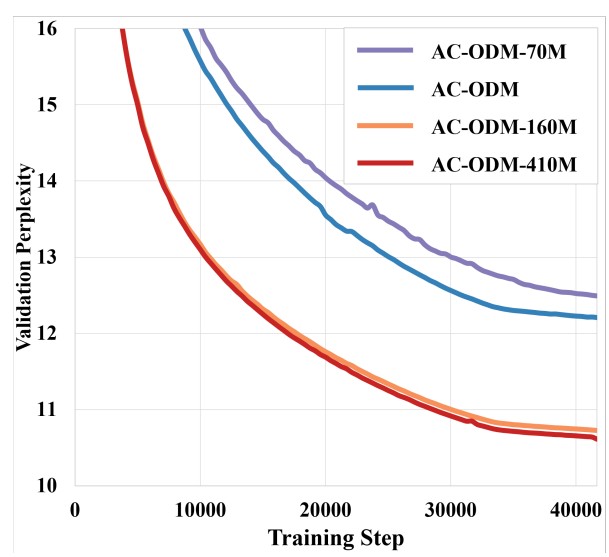

*Figure 7.* Validation perplexity for a 1B target using policies learned with proxy LLMs of different sizes (average over 22 Pile domains).

## F. Evolution of Domain Weights During Training

Figure 8a–8d illustrate the evolution of domain weights across 22 distinct domains in The Pile dataset during training 1B Pythia model under AC-ODM algorithm. AC-ODM initializes from the original domain weights of The Pile and undergoes dynamic updates during the warmup phase. After approximately 15,000 training steps, the domain weights stabilize. Afterward, minor fluctuations are observed, which correspond to the evolving state of the LLM. The adaptive nature of AC-ODM's domain weight generation during this critical phase allows it to better align with the evolving model state, thereby facilitating faster reductions in both training loss and perplexity compared to prior methods.

Both AC-ODM and ODM algorithms eventually converge to stable domain weights. However, AC-ODM exhibits more substantial adjustments in domain weights during the first third of training, while ODM (Albalak et al., 2023) stabilizes after only the first fifth of the total training steps. Notably, even after reaching stability, AC-ODM continues to experience slight fluctuations in domain weights, enabling dynamic adaptation to evolving LLM state. In contrast, domain weights in ODM

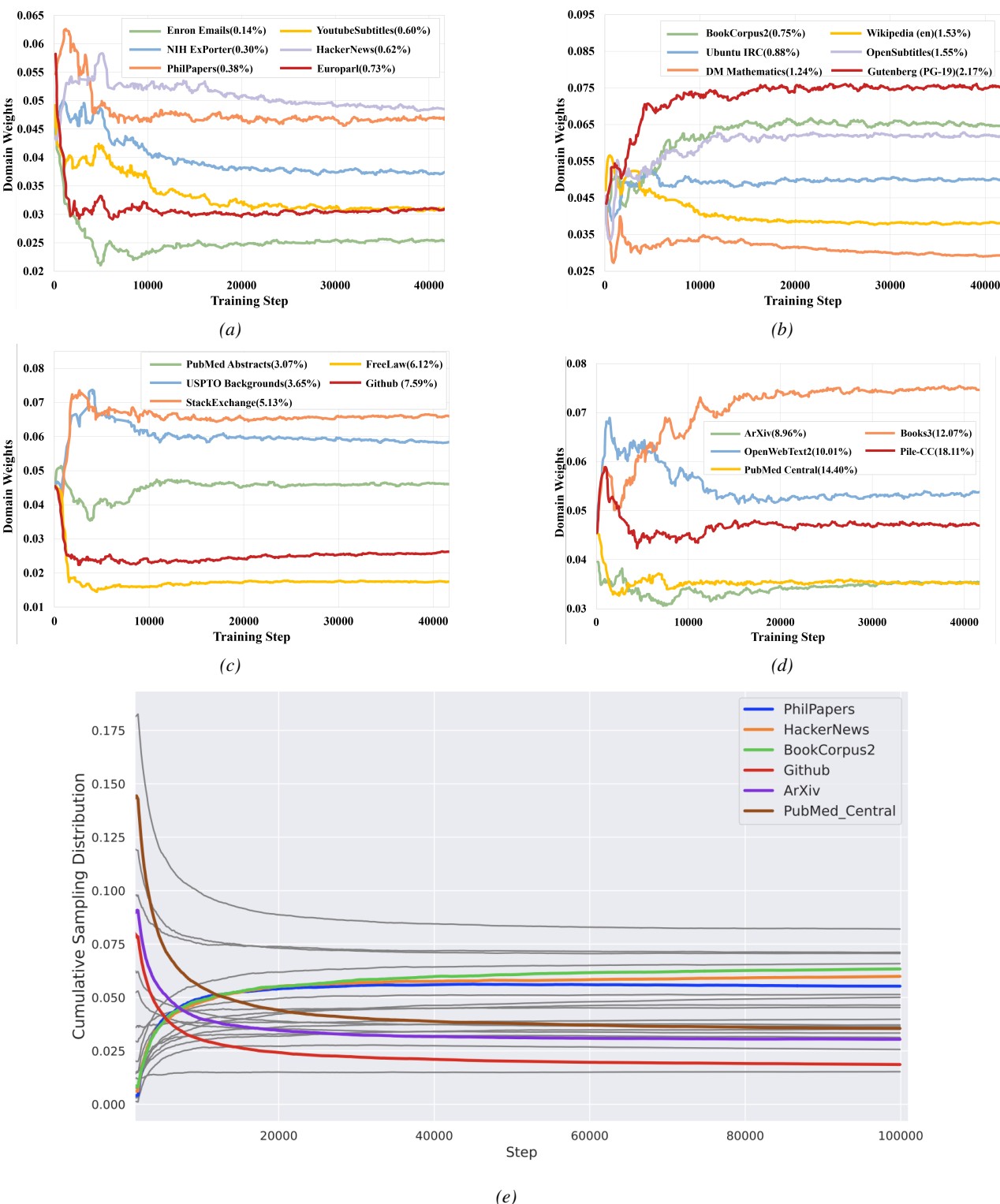

*Figure 8.* Evolution of domain weights during training. The legend indicates the proportion of tokens of each domain (in percentage). (a) Six domains with smallest token proportions; (b) Six domains with token proportions below 3%; (c) Five domains with token proportions below 8%; (d) Five domains with highest token proportions; (e) The cumulative sampling distribution of ODM (Albalak et al., 2023).

remain nearly constant in the later stages of training, indicating a lack of flexibility in response to parameter updates in later stage.

A comparison of domains with the large magnitudes of increases or decreases in weights across Figure 8a–8d reveals consistent patterns. Regardless of the token proportion, domains characterized by high-quality and general-purpose texts tend to experience weight increases during training. Examples include HackerNews in Figure 8a, Gutenberg (PG-19) and BookCorpus2 in Figure 8b, StackExchange and USPTO Backgrounds in Figure 8c, and Book3 in Figure 8d. In contrast, domains containing noisier texts or highly domain-specific contents exhibit significant weight reductions, such as Enron Emails in Figure 8a, DM Mathematics and Wikipedia (en) in Figure 8b, Github and FreeLaw in Figure 8c, and PubMed Central in Figure 8d. These observations align with human intuitive expectations: during LLM pretraining, data domains rich in high-quality, generalizable content are more effective at driving model convergence in the early stages of training.

## G. Analysis of Results of MMLU Tasks

*Table 7.* Zero-shot accuracy of AC-ODMs among different groups in MMLU.

| Algorithm | STEM | Social Sciences | Humanities | Other | Average |
|---|---|---|---|---|---|
| AC-ODM | 0.24213 | 0.30433 | 0.25381 | 0.24626 | 0.25146 |
| AC-ODM-410M | 0.28219 | 0.38231 | 0.29908 | 0.28924 | 0.29980 |

We evaluate the performance of AC-ODM across four domain-specific groups in the MMLU benchmark, along with the overall average accuracy. As shown in Table 7, AC-ODM achieves better accuracy in the *Social Sciences* group, achieving approximately 21% higher than average. This indicates that AC-ODM effectively adapts to domain shifts in this group, likely benefiting from the alignment between Social Sciences content and the training distribution in The Pile. In contrast, AC-ODM underperforms in the *STEM* and *Other* groups, where accuracy falls slightly below the overall average. The *Humanities* group yields performance close to the average. These observations suggest that AC-ODM facilitates the LLM's ability to better acquire and generalize semantic patterns related to humanities and social science domains from The Pile.

Compared to the direct application of AC-ODM, the proxy-based AC-ODM-410M variant consistently improves performance across all groups, yielding an overall 19% increase in average accuracy. The most notable gains occur in the *Social Sciences* and *Other* groups, with improvements of 26% and 17%, respectively. These results indicate that AC-ODM trained on a 410M-parameter proxy model can effectively capture the underlying domain relationships present in The Pile, which are transferable to larger models and particularly beneficial for tasks involving humanities, social sciences, and general knowledge. However, the relatively limited gains in STEM-related domains also suggest that AC-ODM pays less attention to exploring domain-specific features relevant to science and engineering. This limitation may stem from the relatively low proportion of STEM-related content in The Pile dataset itself, which we would like to investigate in the future.

Figure 9 illustrates the task-level accuracy of AC-ODMs across different groups within the MMLU benchmark. In the *STEM* group, AC-ODMs achieve strong performance on tasks such as *Electrical Engineering* and *Computer Security*. Within the *Social Sciences* group, notable improvements are observed in *US Foreign Policy*, *Professional Psychology*, *High School Psychology*, and *Econometrics*. For the *Humanities* group, AC-ODMs perform well on *World Religions*, *Logical Fallacies*, and *Jurisprudence*. In the *Other* group, tasks such as *Marketing*, *Human Aging*, *College Medicine*, and *Clinical Knowledge* benefit significantly from AC-ODMs. These results suggest that AC-ODM's domain weight optimization strategy effectively guides the LLMs to acquire semantic information associated with general-purpose knowledge domains.

Compared to AC-ODM, the proxy-based AC-ODM-410M consistently improves performance across all tasks. Notably, for particularly challenging tasks such as *High School Statistics*, *Elementary Mathematics*, and *Management*, AC-ODM-410M achieves non-zero accuracy where AC-ODM fails completely (0% accuracy). These findings highlight that the use of a well-trained proxy model during training enables AC-ODM to capture meaningful domain relationships, ultimately enhancing LLM performance. Proxy-based training allows the model to better infer the latent structure of domain-specific knowledge while fulfilling difficult tasks, thereby leading to more effective adaptation and improved generalization.

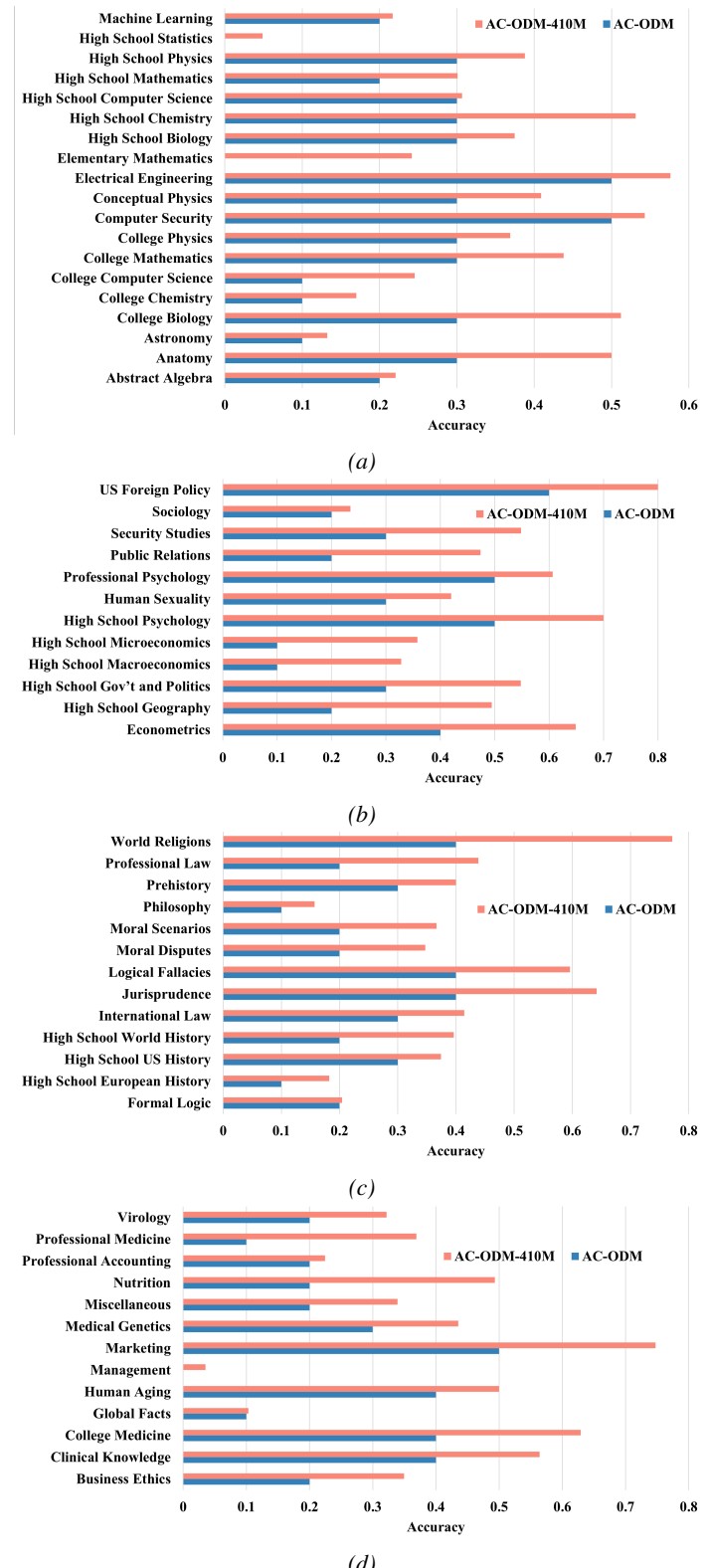

*Figure 9.* Zero-shot accuracy of AC-ODMs across MMLU tasks, grouped by subject category. (a) STEM; (b) Social Sciences; (c) Humanities; (d) Other.

# H. Additional Camera-Ready Experiments

### H.1. Policy Model Size Sensitivity

We vary the actor–critic parameter budget as a fraction of the target LLM size and report final test perplexity on The Pile. Table 8 shows that very small policies underfit the training dynamics, while larger policies provide little additional benefit after roughly $0.25\%$–$0.5\%$ of the target size. This supports the default design choice: AC-ODM needs enough capacity to model cross-domain interactions, but its policy can remain orders of magnitude smaller than the LLM.

*Table 8.* Final test perplexity on The Pile with different policy-model sizes. Percentages denote the policy size relative to the target LLM.

| Target LLM | 0.05% | 0.15% | 0.25% | 0.5% | 0.75% |
|---|---|---|---|---|---|
| 410M | 15.15 | 14.86 | **14.84** | 14.92 | 15.02 |
| 1B | 13.19 | 12.33 | **12.21** | **12.21** | 12.22 |

### H.2. Reward Stabilization

We also track the mean reward over the 22 Pile domains during non-proxy training. As shown in Table 9, the reward rises rapidly and then stabilizes at a high level, broadly matching the stabilization of domain weights observed in Appendix F. The curve is not expected to be monotonic because the reward is computed from stochastic gradients and the model state keeps changing, but its plateau indicates that the learned policy converges to consistently constructive gradient interactions.

*Table 9.* Average reward over 22 Pile domains during non-proxy AC-ODM training.

| Metric | 10,425 | 20,834 | 31,250 | 41,667 |
|---|---|---|---|---|
| Reward | 0.21 | 0.89 | **0.92** | 0.91 |

### H.3. Proxy-Target Scale-Up

To test whether proxy transfer remains useful at a larger scale, we train a Pythia-12B target using a policy learned from a Pythia-1B proxy. Table 10 reports validation perplexity over the first 25B tokens. AC-ODM maintains a clear advantage over ODM throughout training, suggesting that stronger proxies can learn policies that remain transferable to substantially larger targets.

*Table 10.* Validation perplexity on The Pile during Pythia-12B pretraining.

| Method | 5,208 | 10,416 | 15,624 | 20,832 |
|---|---|---|---|---|
| ODM | 14.53 | 10.01 | 8.55 | 7.32 |
| AC-ODM (1B proxy) | **12.87** | **7.56** | **5.91** | **4.24** |

### H.4. Larger LLaMA-Style Models

We further evaluate non-proxy AC-ODM on larger LLaMA-style decoders. The 3B model follows the layer configuration of LLaMA 3.2-text, and the 7B model follows the layer configuration of LLaMA 3. Table 11 shows that validation perplexity improves consistently as model scale increases, confirming that the AC-ODM training recipe remains effective beyond the 0.9B LLaMA-style setting in the main text.

### H.5. Comparison with RegMix

RegMix is a strong static mixture optimization baseline. We include an additional half-budget Pythia-1B comparison under the same training setup. Table 12 shows that AC-ODM improves validation perplexity at every measured checkpoint, reinforcing the main conclusion that adapting mixtures online is more effective than fixing a globally optimized static mixture.

*Table 11.* Validation perplexity of LLaMA-style models on The Pile in non-proxy mode.

| Model size | 5,208 | 10,416 | 15,624 | 20,832 |
|---|---|---|---|---|
| 0.9B | 18.06 | 14.82 | 13.65 | 12.86 |
| 3B | 15.31 | 13.24 | 11.15 | 10.59 |
| 7B | **13.98** | **10.99** | **9.54** | **8.79** |

*Table 12.* Validation perplexity on The Pile during Pythia-1B pretraining.

| Method | 5,208 | 10,416 | 15,624 | 20,832 |
|---|---|---|---|---|
| RegMix | 19.46 | 16.86 | 15.61 | 14.53 |
| AC-ODM | **18.05** | **15.44** | **14.25** | **13.34** |

