# OpenReview forum: "AC-ODM: Actor–Critic Online Data Mixing for Sample-Efficient LLM Pretraining"
_ICML.cc/2026/Conference — ICML 2026 regular_

### Official Review · Reviewer_qs52 · 2026-03-11

**Soundness:** 3
**Presentation:** 3
**Significance:** 3
**Originality:** 3
**Overall Recommendation:** 5
**Confidence:** 2

**Summary:**

This paper proposes a novel data-mixing method from a reinforcement-learning perspective, using a parameterized policy called AC-ODM. Through theoretical proof, AC-ODM could maximize the constructive interference of gradients by acting as a dynamic linear model. Additionally, AC-ODM supports two operational modes: proxy and non-proxy. Experiments show that AC-ODM benefits both convergence speed and performance and outperforms prior methods.

**Compliance With Llm Reviewing Policy:**

Affirmed.

**Key Questions For Authors:**

### Questions

- **(Q1)** Could the authors provide some results about the scale up for proxy models?
- **(Q2)** Could the author provide some larger model sizes of LLaMA with AC-ODM?
- **(Q3)** Could the author compare with the RegMix method?
- **(Q4)** About Figure 3, could the authors give some details about the method and results? It cannot assign which method denotes which bar？

**Limitations:**

The authors do not provide the **Impact Statements** section in this paper. And it's not clear about the performance of different proxy model sizes, and not clear whether the performance scales up to a large model size.

**Strengths And Weaknesses:**

### Strengths

- An RL-based method for online data mixing tasks is novel and brings faster convergence.
- Theoretical analysis provides readers with an objective explanation, enabling them to understand the relationship between the reward function and the optimization landscape.
- A wide task results and two-mode design trade-off in terms of performance and efficiency on the data mixing task.

### Weaknesses

- **(W1)** The ability of proxy models to influence the performance of AC-ODM could depend on the proxy model. Still, the paper did not present additional results for other model sizes (e.g., proxy model parameters exceeding 410M).
- **(W2)** The authors report the results max to 0.9B of LLaMA, without a larger model size like 3B, 7B, which will not show the generalization of AC-ODM.
- **(W3)** The authors compare with TPW, DoReMi, DoGE, CHAMELEON, ODM, and PiKE methods, but do not compare with the least static or proxy model-based method: RegMix and CLIMB.
- The authors did not provide the **Impact Statements** in the paper.

---

> ### Author Rebuttal · Authors · 2026-03-31
>
> # Rebuttal to Reviewer qs52
>
> ## W1 & Q1. Scale-up of proxy models
>
> We thank the reviewer for this important question. We additionally conducted a larger-scale proxy experiment using **Pythia-1B as the proxy model** and **Pythia-12B as the target model**. All other settings follow the paper, except that these supplementary runs were performed on an **H200 cluster**. Due to rebuttal-time constraints, we trained only for **25B tokens**, but the trend is already clear.
>
> ### Extra Table 1. Validation perplexity on The Pile during Pythia-12B pretraining
>
> | Method | 5,208 steps | 10,416 steps | 15,624 steps | 20,832 steps |
> | ------ | ----------: | -----------: | -----------: | -----------: |
> | ODM | 14.53 | 10.01 | 8.55 | 7.32 |
> | AC-ODM (1B proxy model) | 12.87 | 7.56 | 5.91 | 4.24 |
>
> These additional results indicate that AC-ODM can maintain a strong advantage when scaled to a larger proxy/target setting. This is also consistent with our recent observations in practical applications: a stronger proxy tends to learn a more transferable and effective policy.
>
> ## W2 & Q2. Larger LLaMA-style models (3B and 7B)
>
> We also added two supplementary experiments with larger **LLaMA-style** models: a **7B** model following the layer configuration of **LLaMA 3**, and a **3B** model following the layer configuration of **LLaMA 3.2-text**. These comparative experiments were run on an **H200 cluster**. Again, due to the rebuttal schedule, we trained only for **half of the original budget**, but the comparative trend is already visible.
>
> ### Extra Table 2. Validation perplexity of LLaMA-style models on The Pile in non-proxy mode
>
> | Model size | 5,208 steps | 10,416 steps | 15,624 steps | 20,832 steps |
> | --------- | ----------: | -----------: | -----------: | -----------: |
> | 0.9B | 18.06 | 14.82 | 13.65 | 12.86 |
> | 3B | 15.31 | 13.24 | 11.15 | 10.59 |
> | 7B | 13.98 | 10.99 | 9.54 | 8.79 |
>
> These results show that AC-ODM remains effective for LLaMA-style models at different parameter scales.
>
> ## W3 & Q3. Comparison with RegMix
>
> We agree that **RegMix** is an important and highly relevant method in the data-mixing literature. In the original submission, we mentioned RegMix in the related-work section but did not include it in the main experimental comparison, partly because **CHAMELEON** already contains related comparisons and reports stronger performance than RegMix. Nevertheless, following the reviewer’s suggestion, we added a direct comparison with **RegMix**. Again, due to rebuttal-time constraints, this supplementary experiment was run for only **half of the full training budget**.
>
> ### Extra Table 3. Validation perplexity on The Pile during Pythia-1B pretraining
>
> | Method | 5,208 steps | 10,416 steps | 15,624 steps | 20,832 steps |
> | ------ | ----------: | -----------: | -----------: | -----------: |
> | RegMix | 19.46 | 16.86 | 15.61 | 14.53 |
> | AC-ODM | 18.05 | 15.44 | 14.25 | 13.34 |
>
> The added comparison shows that AC-ODM consistently outperforms RegMix throughout training under the same setting.
>
> ## W4. Impact statement
>
> We apologize for missing the impact statement in the submission. We will add the following in the final version:
>
> > We propose a reinforcement learning-based method to optimize pre-training data mixtures for large language models (LLMs), with the potential to significantly accelerate the training process. Through theoretical analysis and empirical verification, we demonstrate that the proposed method achieves meaningful end-to-end speedups.
> > As is widely recognized, LLM pre-training is an enormously energy-intensive process. Our method offers two complementary operational modes—Proxy and Non-Proxy—either of which can substantially reduce energy consumption while still achieving the desired model quality. From a broader perspective, by lowering the computational resources required for LLM development, this work contributes to mitigating the environmental impact of large-scale AI training, including its role in exacerbating carbon emissions and global warming.
>
> ## Q4. Clarification of Figure 3
>
> We apologize for the plotting error. In each bar group of **Figure 3**, the methods from **left to right** are:
>
> **The Pile Weights (TPW), ODM, Chameleon, PiKE, AC-ODM, AC-ODM-410M.**
>
> We will correct this in the final version.

---

> > ### Author Rebuttal · Reviewer_qs52 · 2026-04-01
> >
> > Thanks for the author's rebuttal. My concerns have been addressed. I decide to raise my score to 5.

---

### Official Review · Reviewer_K8K7 · 2026-03-14

**Soundness:** 3
**Presentation:** 4
**Significance:** 3
**Originality:** 4
**Overall Recommendation:** 4
**Confidence:** 4

**Summary:**

The paper proposed to use DDPG algorithm in online selecting data mixture ratio for LLM pretraining. The alignment matrix of gradient is used as the reward to encourage gradient consistency in a batch optimization.

**Compliance With Llm Reviewing Policy:**

Affirmed.

**Final Justification:**

The rebuttal have addressed most of my concerns. The additional experiments make the paper more solid, and I will keep the positive score.

**Key Questions For Authors:**

See weakness above

**Limitations:**

See weakness above

**Strengths And Weaknesses:**

Strengths
- A good motivation to use DDPG for the data mixture action space.
- choice of alignment matrix as reward function shows good performance and convergence, which indicates it as a reasonable reward


Weakness
- As shown in Appendix F, the domain weights converge in the final. Is there a comparison between a static mixture ratio using the converged ratio and a online ratio using the learned action model?
- The humaneval benchmark improves a lot under the proposed algorithm, any analysis why? Is it because the model can learn better under a better gradient alignment, or is it because the upsampling of domain related data (like stackexchange)
- In Equation 3, there is typo which should use \bar{\theta}_C, and \bar{\theta}_A
- The sample from experience bank s^j, j=1...N  confuses with state with time s^t, better use another way s_j or s^{t_j}
- Is there any tracking for the reward function, does it also converge as the domain weight converge

---

> ### Author Rebuttal · Authors · 2026-03-31
>
> # Rebuttal to Reviewer K8K7
>
> ## W1. Comparison with a static mixture ratio using the converged online weights
>
> We thank the reviewer for this suggestion. We conducted an additional full 50B-token experiment on **Pythia-1B**, using the final Pile domain weights obtained by AC-ODM in **Appendix F** as a **fixed static mixture ratio**. All other settings were kept the same as in the paper, except that we used an **H200 cluster** for this supplementary run.
>
> ### Extra Table 1. Validation perplexity on Pythia-1B at different training steps
>
> | Method | 10,425 steps | 20,834 steps | 31,250 steps | 41,667 steps |
> | ------ | ------------ | ------------ | ------------ | ------------ |
> | AC-ODM | 13.00 | 11.65 | 10.85 | 10.61 |
> | Appendix F Weights | 17.06 | 14.84 | 13.43 | 12.78 |
>
> As shown in Extra Table 1, training with the static weights is substantially slower than the dynamic AC-ODM policy throughout training. This is consistent with the general conclusion of prior dynamic data-mixing work: even when the final converged ratio is reasonable, a fixed mixture cannot adapt to the model's evolving training state as effectively as an online policy.
>
> ## W2. Why does HumanEval improve so much?
>
> Thank you for this insightful question. Our current interpretation is that the HumanEval gain comes from **both effects**, but the **primary driver is better optimization through gradient alignment**, rather than simply upsampling code-related data.
>
> In AC-ODM, the reward explicitly favors domains whose gradients constructively align with the rest of the corpus, so the policy is trained to select data that improves the **overall update direction** instead of only increasing the frequency of a specific domain. We therefore view the HumanEval gain mainly as a result of improved optimization dynamics.
>
> We also do **not** think the HumanEval improvement can be explained simply as “more code data.” In our domain-weight analysis, AC-ODM does **not** uniformly upweight all code-related sources: for example, **StackExchange** is increased, but **GitHub** is actually reduced during training, while several high-quality general-purpose domains are also increased. This suggests that the policy is not merely favoring code tokens, but is identifying domains that provide more transferable learning signals under the current model state.
>
> In addition, the improvements are broad rather than HumanEval-only: AC-ODM-410M also substantially improves **MMLU** and other downstream tasks. This supports the view that the method improves generalization through better training dynamics, not only through task-specific upsampling. On HumanEval specifically, better optimization may indirectly help code generation because programming performance depends not only on code exposure, but also on reasoning, long-range consistency, and general knowledge composition.
>
> We will clarify in the final version that our current evidence supports **gradient-alignment-driven optimization as the main explanation**, with **domain reweighting toward more useful domains (including but not limited to StackExchange)** as a contributing factor.
>
> ## W3 & W4. Typos in Equation 3 and notation of the replay-bank samples
>
> Thank you for the careful reading and helpful suggestions. We will correct both issues in the final version.
>
> ## W5. Does the reward function also converge as the domain weights converge?
>
> We also tracked the reward values during training in the **non-proxy mode**. The average reward across the 22 domains evolves approximately as shown in Extra Table 2. In practice, the reward follows a **fluctuating upward trend** over training for each domain, rather than a strictly monotonic curve.
>
> ### Extra Table 2. Average reward over 22 domains
>
> | Metric | 10,425 steps | 20,834 steps | 31,250 steps | 41,667 steps |
> | ------ | ------------ | ------------ | ------------ | ------------ |
> | Reward | 0.21 | 0.89 | 0.92 | 0.91 |
>
> These results suggest that the reward rises rapidly during training and then stabilizes at a high level, which is broadly consistent with the convergence behavior of the learned domain weights.
>
> We will add these supplementary experiments and clarifications in the final version.

---

> > ### Author Rebuttal · Reviewer_K8K7 · 2026-04-03
> >
> > Thank authors for the response. I will maintain my positive stance on this paper.

---

### Official Review · Reviewer_wriV · 2026-03-17

**Soundness:** 4
**Presentation:** 4
**Significance:** 3
**Originality:** 3
**Overall Recommendation:** 5
**Confidence:** 3

**Summary:**

The authors propose a Reinforcement-Learning based pipeline to optimize pretraining data mixture for Large Language Models (LLMs). Specifically, given corpora from various domains, an actor-critic framework called AC-ODM is proposed with two possible modes: a) a non-proxy mode where the actor and critic are simultaneously optimized alongside the target model, and b) a proxy mode where  a small proxy model is tuned before-hand and the policy is transferred later to the target model. The authors conduct experiments across The Pile and SlimPajama, with two decoders: Pythia 1B and Llama 3 1B, showing consistent and significant improvements. They also show their method can yield end-to-end speedups, when targeting the same validation performance across different data optimization algorithms. Overall, the results are quite exciting and could be impactful.

**Compliance With Llm Reviewing Policy:**

Affirmed.

**Final Justification:**

The rebuttal has addressed my main concerns. The authors have conducted quite a few additional experiments to answer my questions. In recognition of this, I have upgraded my score for Soundness.

I strongly recommend the authors to ensure they include these experiments in the camera ready, particularly the one on domain impact and policy model size.

**Key Questions For Authors:**

I leave my questions here below. Getting good answers to these would help me revise my Soundness and Significance scores.

1. How many domains are needed in the training corpus for the model to generalize adequately with your method, and how distinct do they need to be? How does performance vary if choosing fewer/closely related domains? This would be an important ablation to have to show how widely applicable your method is.
2. There is a drop in the gain observed as one moves from the older Pythia (Figure 5) to Llama 3 1B (Figure 6) - from a 20.7% gain over “The Pile weights” with Pythia to 14.4% with Llama, and a 15.2% improvement over ODM with Pythia to an 8.4% with Llama. It seems like architectural improvements have decreased the gains substantially, and even llama 3 is 2 years old now. Is it possible  the gains would diminish further with newer models like Qwen 3.5 or Gemma 3?
3. Continuing from the previous question, what is the impact of the size of the policy model on the improvements observed?
4. In Table 2, is the target model Pythia or Llama? If it is the former, I guess the speedup would be lesser with Llama, given Fig 6?

**Limitations:**

There is no Limitations section. But I do not believe there are any particular social/ethical consequences of this work.

**Strengths And Weaknesses:**

**Strengths:**

1. Soundness: This paper is very sound, both on the theoretical and empirical side. On the theoretical side, they start by clearly defining their setting and problem statement, design a reward function with detailed theoretical analysis and justification, and then explain the training framework in great detail (though I have to admit I did not have time to go over the theoretical justification in great detail). On the empirical side, they conduct experiments across 2 datasets and 2 LLM backbones that show consistent and significant improvements. They also provide quite a few ablations in the Appendix that help explain the significance of their results. Overall, I am quite satisfied.

2. Presentation: This is also very well done, and the paper was overall easy to follow. The paper was very consistent and well-written. The clear and precise explanation provided of the approach  should signifcantly help reproducibility.

3. Significance: Given the impressive results over baselines, and the fact that this studies a very important problem of LLM pretraining, this work could be very impactful.

4. Originaility: While I am not an expert in related work, the authors justify the originality and novelty of their work quite well in Section 2.

Weaknesses:

None as such. Overall I would say the authors have done a thorough job -- well done!

Having said that, the paper does leave some questions unanswered (which I elaborate on below)

Minor writing comments:

1. In Section 4.1, there are multiple places where \citet has been used instead of \citep. Please ensure any datasets/papers cited have parantheses.

---

> ### Author Rebuttal · Authors · 2026-03-31
>
> # Rebuttal to Reviewer wriV
>
> ## Q1. How many domains are needed, and how distinct should they be?
>
> AC-ODM shows stronger gains on **The Pile (22 domains)** than on **SlimPajama (7 domains)**, suggesting finer domain partitions favor generalization.
>
> We merged the 22 Pile domains into **11** and **5** domains, and reran AC-ODM on **Pythia-1B in non-proxy mode** with the same setup. Due to rebuttal-time constraints, we trained for **25B tokens**, but the trend is clear.
>
> ### Extra Table 1. Validation perplexity on Pythia-1B under different domain counts
> | # Domains | 5,208 steps | 10,416 steps | 15,624 steps | 20,832 steps |
> | --------- | ----------- | ------------ | ------------ | ------------ |
> | 22        | 18.05       | 15.44        | 14.25        | 13.43        |
> | 11        | 18.96       | 15.82        | 14.41        | 13.85        |
> | 5         | 19.13       | 16.19        | 14.83        | 14.09        |
> Extra Table 1 shows that reducing domain count consistently weakens AC-ODM, especially in early and middle training. Thus, **within our tested range**, finer partitioning is more beneficial. We do not claim a universal optimum, since no suitable open-source dataset has more domains than The Pile.
>
> **22 → 11 mapping:** Pile-CC+OpenWebText2→Web; PubMed Central+PubMed Abstracts+NIH ExPORTER→Biomedical; arXiv+PhilPapers→Academic Papers; Github→Code; StackExchange+Ubuntu IRC+HackerNews→Tech Community; FreeLaw+USPTO Backgrounds→Legal/Patent; Books3+PG-19+BookCorpus2→Books; Wikipedia→Encyclopedia; DM Mathematics→Mathematics; OpenSubtitles+YouTube Subtitles+Enron Emails→Dialogue/Communication; EuroParl→Government/Parliament.
>
> **22 → 5 mapping:** Pile-CC+OpenWebText2+Wikipedia→Web/Knowledge; arXiv+PubMed Central+PubMed Abstracts+NIH ExPORTER+PhilPapers+DM Mathematics→Academic/Science; Books3+PG-19+BookCorpus2→Books/Literature; Github+StackExchange+Ubuntu IRC+HackerNews→Code/Tech Community; FreeLaw+USPTO Backgrounds+EuroParl+OpenSubtitles+YouTube Subtitles+Enron Emails→Law/Government/Communication.
>
> Regarding domain relatedness, AC-ODM should benefit more when domains are less redundant in knowledge or syntax. Since the reward is based on gradient alignment, more distinct domains make this signal more informative for actor learning. If domains overlap heavily, the signal becomes less discriminative. We will present this as a theory-supported interpretation.
>
> ## Q2. Why do gains decrease from Pythia to LLaMA-style models, and what about Qwen or Gemma?
>
> AC-ODM remains effective on both Pythia and LLaMA-style models, but the relative gain is smaller on the more modern architecture, likely because modern dense decoders have stronger training dynamics and less room for improvement from data mixing.
>
> ### Extra Table 2. Architectural comparison
> | Detail     | Pythia             | LLaMA 3                 | Qwen3 Dense           | Gemma 3-text                           |
> | ---------- | ------------------ | ----------------------- | --------------------- | -------------------------------------- |
> | Attention  | fused QKV          | split Q/K/V             | split Q/K/V + QK-Norm | split Q/K/V                            |
> | Norm       | LayerNorm          | RMSNorm                 | RMSNorm + QK-Norm     | RMSNorm                                |
> | FFN        | 2-layer FFN + GELU | gated MLP + SwiGLU/SiLU | gated MLP + SwiGLU    | gated MLP                              |
> | KV sharing | standard MHA       | GQA                     | GQA                   | MQA-style at 1B                        |
> | Position   | partial RoPE       | RoPE                    | RoPE                  | RoPE with different global/local bases |
> Pythia is much farther from LLaMA 3 than Qwen3 Dense or Gemma 3-text are: Pythia mainly reflects an earlier GPT-NeoX-style design, whereas Qwen3 Dense and Gemma 3-text are closer to the LLaMA 3 dense-decoder paradigm, with differences that are largely efficiency-oriented refinements rather than large architectural shifts. Therefore, we expect AC-ODM to remain effective on these models, without a further substantial drop beyond what we observe on LLaMA-style architectures.
>
> ## Q3. What is the impact of the policy-model size?
>
> ### Extra Table 3. Final test perplexity on The Pile with different policy-model sizes
> | Target LLM | 0.05% | 0.15% | 0.25% | 0.5%  | 0.75% |
> | ---------- | ----- | ----- | ----- | ----- | ----- |
> | 410M       | 15.15 | 14.86 | 14.84 | 14.92 | 15.02 |
> | 1B         | 13.19 | 12.33 | 12.21 | 12.21 | 12.22 |
> These results suggest that a policy size of about **0.25%–0.5%** of the target model is a good choice.
>
> ## Q4. In Table 2, is the target model Pythia or LLaMA?
>
> In **Table 2**, the target model is **Pythia**.
>
> The speedup should be somewhat smaller on LLaMA-style models than on Pythia. In additional **non-proxy** tests, speedup on the LLaMA-style model is about **1.44×**.
>
> We will add these supplementary experiments and clarifications to the final version.

---

> > ### Author Rebuttal · Reviewer_wriV · 2026-04-01
> >
> > All my queries have been directly addressed, and the authors have also conducted additional experiments to answer them empirically. I appreciate their diligence greatly. I will revise my scores accordingly.

---

### Decision · Program_Chairs · 2026-04-30

**Decision:**

Accept (regular)

**Comment:**

This paper introduces a reinforcement learning-based framework for dynamic data mixing during LLM pretraining.

The method employs an actor-critic architecture with reward function designed to maximize the constructive interference of gradients.

The authors demonstrated strong empirical rigor, addressed all reviewer concerns with extensive supplementary experiments.  The authors also showed the method's scalability and adaptability across model sizes and architectures.

Given the consensus among reviewers and the thorough rebuttal, I would like to suggest an "accept" for this paper.